# Survival-Related Genes on Chromosomes 6 and 17 in Medulloblastoma

**DOI:** 10.3390/ijms25147506

**Published:** 2024-07-09

**Authors:** Jerry Vriend, Xiao-Qing Liu

**Affiliations:** 1Department of Human Anatomy and Cell Science, University of Manitoba, Winnipeg, MB R3T 2N2, Canada; 2Department of Obstetrics, Gynecology and Reproductive Sciences, University of Manitoba, Winnipeg, MB R3T 2N2, Canada; xiaoqing.liu@umanitoba.ca; 3Biochemistry and Medical Genetics, Rady Faculty of Health Sciences, University of Manitoba, Winnipeg, MB R3T 2N2, Canada

**Keywords:** medulloblastoma, gene expression, survival-related genes, hazard ratios, chromosome 6, chromosome 17, *HMGA1*

## Abstract

Survival of Medulloblastoma (MB) depends on various factors, including the gene expression profiles of MB tumor tissues. In this study, we identified 967 MB survival-related genes (SRGs) using a gene expression dataset and the Cox proportional hazards regression model. Notably, the SRGs were over-represented on chromosomes 6 and 17, known for the abnormalities monosomy 6 and isochromosome 17 in MB. The most significant SRG was *HMGA1* (high mobility group AT-hook 1) on chromosome 6, which is a known oncogene and a histone H1 competitor. High expression of *HMGA1* was associated with worse survival, primarily in the Group 3γ subtype. The high expression of *HMGA1* was unrelated to any known somatic copy number alteration. Most SRGs on chromosome 17p were associated with low expression in Group 4β, the MB subtype, with 93% deletion of 17p and 98% copy gain of 17q. GO enrichment analysis showed that both chromosomes 6 and 17 included SRGs related to telomere maintenance and provided a rationale for testing telomerase inhibitors in Group 3 MBs. We conclude that *HMGA1*, along with other SRGs on chromosomes 6 and 17, warrant further investigation as potential therapeutic targets in selected subgroups or subtypes of MB.

## 1. Introduction

Gene expression in medulloblastoma (MB) has been studied in terms of major histopathological types as well as molecular subgroups [1]. These studies have generated large publicly available datasets that can be used to mine information on genes that can potentially be targeted for therapeutic purposes. In 2012, a consensus study from a number of laboratories identified four major molecular subgroups of MB: Group 3; Group 4; SHH (Sonic Hedgehog); and WNT (Wingless) [2]. While the SHH signaling pathway is activated in the SHH MB group and the WNT signaling pathway is activated in the WNT MB group, no single signaling pathway has been identified as characteristically activated in either Group 3 or Group 4 MBs. In 2017, Cavalli et al. published a landmark study [3] that confirmed the four major molecular subgroups of MB and defined subtypes within each subgroup. In 2019, Weishaupt et al. integrated and normalized 23 transcriptome datasets, which allowed for the comparison of the gene expression values in MB subgroups to the values from a non-tumor group [4].

Several chromosomal aberrations have been related to MB. Thompson et al. (2006) [1] noted that downregulated genes on chromosome 6 were over-represented among all downregulated genes in MB and predicted a loss of a copy of chromosome 6 in the WNT group of MB. These findings were further confirmed by fluorescence in situ hybridization (FISH) analysis [1] and replicated by subsequent studies [5,6]. Cavalli et al. (2017) reported monosomy of chromosome 6 in 48 of the 49 subjects from the WNTα subtype [3].

Thompson et al. (2006) also detected deletions of chromosome 17p in one of their MB groups in the context of isochromosome 17 for most cases [1]. Previous studies have also reported isochromosome 17 in MB Groups 3 and 4 [2,3,7]. Other chromosomal aberrations in MB include deletions of chromosomes 9 and 10 in the SHH group [2], isolated deletions of 17p [8], and copy number gains of 17p in MB cell lines [9]. These chromosomal aberrations have been noted in various reports on MB subgroups, including the report of Cavalli et al. [3].

The report by Cavalli et al. [3] included survival curves for each MB subgroup and subtype. Zhu et al. [10], on the other hand, identified a 12-gene signature independent of MB subgroups and subtypes to predict survival.

Herein, we examined the publicly available MB gene expression data to identify genes significantly associated with overall survival using the Cox proportional hazards regression model. We found that the survival-related genes (SRGs) are over-represented on chromosomes 6 and 17. Second, we provide detailed gene expression profiles of these SRGs on these two chromosomes and relate them to the MB subgroups and subtypes described by Cavalli et al. (2017) [3], where possible. Furthermore, we relate the transcriptome findings to the literature on monosomy 6 and isochromosome 17 in MB [1,11,12]. Finally, we provide information on the major biological processes represented by the SRGs on these chromosomes and suggest several potential therapeutic targets. This is the first comprehensive analysis of SRGs on chromosomes 6 and 17.

## 2. Results

Of the 763 samples from the MB dataset by Cavalli et al. [3], 467 were labeled as alive, 165 as not alive, and 131 had missing survival information. Figure 1 illustrates the well-known pattern of survival in the four consensus MB subgroups [2,13]. Individuals from the WNT subgroup had the best survival rate, while those from Group 3 had the worst survival, with intermediate survival rates in Group 4 and the SHH group. Within each subgroup, the survival outcomes were also different among the 12 MB subtypes but statistically not as significant as the differences in survival rates among the four MB subgroups (Appendix A).

Our analysis revealed that age was significantly associated with survival years (*p* = 0.02) but not overall survival status (*p* = 0.6). Sex was not significantly associated with either survival years (*p* = 0.2) (Appendix A) or overall survival status (*p* = 0.55). Based on these findings, we selected age as a covariate in our survival analysis model.

A total of 967 SRGs were identified with the Benjamini–Hochberg *p*-value < 0.05 after adjusting for age. Appendix A lists the survival analysis results of the SRGs by chromosome using the Cavalli dataset.

GO enrichment analysis showed that a total of 95 Gene Ontology terms in 16 GO groups were statistically associated (*p* < 0.01) with our list of SRGs (Appendix A). The percentage of GO terms per group for all SRGs is shown in Appendix A. The highest percentage of GO terms per group was in the GO group “regulation of chromosome organization”. The SRGs contributing to this GO group in the Cavalli dataset included five genes coding for proteins making up the TCP1 ring complex (*TCP1*, *CCT2*, *CCT3*, *CCT4*, *CCT8*) in each of the 19 GO terms of the group. The GO terms comprising this group were related to the regulation of the telomere and telomerase RNA localization to the Cajal body (Appendix A). The next most significant GO group represented in the list of SRGs was the mitotic cell cycle process. Of the 967 SRGs, 51 were on the Ongene list of human oncogenes.

Compared to other chromosomes, SRGs on chromosomes 6 and 17 were significantly over-represented among the 967 SRGs, with 9.73% and 9.86% on chromosomes 6 and 17, respectively, versus an average of 5.89% on the other chromosomes (*p* < 0.0001) (Figure 2A). Our analysis focuses on the SRGs located on these two chromosomes. SRG hazard ratios by chromosome are shown in Figure 2B.

### 2.1. SRGs on Chromosome 6

We observed decreased expression of many SRGs on chromosome 6 in the WNTα subtype, as expected for monosomy 6 in this subgroup. However, we also observed some major unexpected findings of SRGs on chromosome 6. Figure 3 shows the GO processes associated with SRGs on chromosome 6. The greatest number of GO terms was associated with nucleosome assembly. Below, we present the findings separately for the short and the long arms of chromosome 6 (6p and 6q).

#### 2.1.1. Chromosome 6p and Expression of SRGs

The expression of SRGs on chromosome 6p is presented in the heatmap of Figure 4. Sixty-four of the 102 SRGs on chromosome 6 are located on its short arm (6p). For 61 of the 64 SRGs on 6p, high expression was associated with worse survival (HR > 1). For the majority of these genes, high expression (including higher expression than the NT group in the Swartling dataset) was found in the Group 3γ subjects, the subtype with the worst prognosis, while low expression was noted in many of the WNTα subjects (Figure 4), the subtype with monosomy 6 and the best prognosis. However, several genes located on the histone cluster at 6p22.1–22.2 showed high expression (Figure 4) in the WNT group (including higher than the NT group in the Swartling dataset) and had poor prognosis (HR > 1).

As noted, GO analysis of chromosome 6 SRGs showed enrichment of the genes involved in nucleosome assembly (Figure 3). The SRGs were *BRD2*, *CENPW*, *DAXX*, *H1-2*, *H2BC11*, *H2BC15*, *H2BC17*, *H2BC4*, *H2BC5*, *H2BC8*, *H3C2*, and *HMGA1*. These genes, with the exception of *CENPW*, are located at 6p.

The group of nucleosome-related genes consisted of four SRGs coding core nucleosomal histone components (*H2BC11*, *H2BC17, H2BC5*, *H3C2*) and *H1.2*, which encodes a histone linker. These genes are all located in the histone cluster at 6p22.1 and 6p22.2 [14] and cluster together in the heatmap of 6p (Figure 4). Additional 6p SRGs related to nucleosomes include *DAXX*, which codes for a histone linker; *BRD2*, which encodes a histone transcription regulator; and *HMGA1*, a histone H1 competitor [15]. The latter three genes are located at 6p21.31, 6p21.32, and 6p21.32, respectively. Expression of *HMGA1* was specifically elevated in Group 3γ MB in the Cavalli dataset.

Furthermore, *HMGA1* was the most significant SRG of all the 967 SRGs (HR = 1.68, *p* = 8.36 × 10^−13^). *HMGA1*, located at 6p21.31, encodes the high-mobility group AT-Hook protein I. In the Weishaupt et al. (2019) [4] dataset, the expression of *HMGA1* was significantly higher in the Group 3 subjects than in the subjects from the other groups, including the non-tumor group. The *HMGA1* expression was specifically elevated in the Group 3γ subjects, 2-fold greater than its expression in the subjects from the other subtypes (Figure 5A). The Group 3γ subtype (n = 40) had poor survival. Figure 5B shows that high expression of *HMGA1* was associated with poor survival.

Secondary analysis of individuals with no known somatic copy number alterations (SCNAs) of chromosome 6 showed that expression of the 6p SRGs, *H2BC11*, *H2BC4*, *H2BC8*, and *HMGA1*, was not associated with monosomy 6 or any other known SCNA.

#### 2.1.2. Chromosome 6q SRGs

Figure 6 shows the heatmap of the gene expression of the 38 SRGs on chromosome 6q in the Cavalli et al. dataset. For 33 of the 38 SRGs on chromosome 6q, high gene expression was associated with poor survival (HR > 1). High expression for most of the 6q SRGs was noted in Group 3, while lower expression of most of the 6q SRGs was found in the WNTα subtype (Figure 6). SRGs with high expression in Group 3γ (and worse survival) included *ADGRG6* (*GPR126*), *NUS1*, *MDN1*, *SYNCRIP*, *WASF1*, and *LIN28B*. *SYNCRIP* is on the list of driver genes of Northcott [16].

*CENPW*, located on 6q, was shown to contribute to the nucleosome assembly biological process, along with several SRGs of 6p (see above). *CENPW* codes for a protein that binds to nucleosomes at the centromere. *CENPW* expression was highest in Group 4α (Figure 7A). Worse survival was associated with high expression of *CENPW* (Figure 7B).

Chromosome 6q also contributed two genes, *TCP1* and *MAP3K4*, to the most significant GO term over-represented in all 963 SRGs: regulation of chromosome organization. *TCP1* (aka *CCT1*) encodes a component of the TCP1 ring complex, a structure that assists in protein folding in cells. *MAP3K4* encodes a protein kinase. Kaplan–Meier analysis showed that high expression of *TCP1* was associated with poor survival (*p* = 5.19 × 10^−10^).

### 2.2. SRGs on Chromosome 17

Figure 8 shows the percentage of GO terms per group associated with SRGs on chromosome 17. The greatest number of GO terms was associated with membrane disassembly. When individuals with SCNAs were removed from the Cox proportional hazards regression analysis there were no longer any significant chromosome 17 SRGs. This would suggest the probability that SRGs on chromosome 17 are related to copy number variations, including isochromosome 17, or a small sample size after the exclusion. According to the supplementary data in Cavalli et al. [3], SCNAs of 17p and 17q were found in 403 out of 763 individuals in the study, mostly in MB Groups 3 and 4. Herein, we present the findings separately for the short and the long arms of chromosomes 17 (17p and 17q).

#### 2.2.1. Chromosome 17p SRGs

Seventy-two of 74 SRGs on 17p showed that low gene expression was associated with poor survival, and high expression was associated with survival protection (HR < 1). Figure 9 illustrates the heatmap of the gene expression levels for the 17p SRGs by MB subtype. The most significant SRGs had low expression in Group 4β (Figure 9). Cavalli et al. (2017) found that 87 of the 109 subjects (80%) in this subtype Group 4β had isochromosome 17 [3]. In their supplementary table of SCNAs (Cavalli et al. (2017) Appendix A), however, we find that 93% of Group 4β had a deletion of 17p. Two of the most significant 17p SRGs are *SCO1* (HR = 0.73, *p* = 2.21 × 10^−5^) and *TTC19* (HR = 0.68, *p* = 4.52 × 10^−7^). Low expression of *SCO1* and *TTC19* in Group 4β and significantly worse survival with low expression were noted compared to the other MB subtypes.

#### 2.2.2. Chromosome 17 and Telomere Genes

In our analysis of the SRGs on chromosome 17, the most frequently represented cytogenetic band on chromosome 17p was 17p13.1, with 27 SRGs (Figure 10A). Three of the 17p SRGs, *SMG6* (at 17p13.3), *RPA1* (also at 17p13.3), and *CTC1* (at 17p13.1), contribute to telomere regulation.

#### 2.2.3. GO Analysis of Chromosome 17p SRGs

The pathway analysis results of 17p SRGs are shown in Table 1. It shows that *CTDNEP1*, a driver gene [16], along with the SRGs *NDEL1* and *PAFAH1B1*, contributed to the most significantly enriched GO term nuclear membrane disassembly (Table 1). The SRGs, *SCO1*, and *TTC19*, contributed to the GO process of cytochrome complex assembly.

#### 2.2.4. Chromosome 17q

The expression of the 17q SRGs by subgroup and subtype is illustrated in Figure 11. As shown in the heatmap, a group of seven, SRGs, *MIR183A*, *P3H4*, *FZD2*, *SLFN11*, *ZNF385C*, *NXPH3*, and *AXIN2*, was over-expressed in the WNT subgroup. High expression was associated with survival protection (HR < 1).

On the other hand, for 28 out of the 45 SRGs on chromosome 17q, high expression was associated with worse survival (HR > 1) (Figure 11). All twelve SRGs at 17q25.3 (*SEC14L1*, *FASN*, *DUS1L*, *MRPL12*, *PCYT2*, *NPTX1*, *ANAPC11*, *EIF4A3*, *RAC3*, *MCRIP1*, *ALYREF*, *CCDC137*) showed high expression related to worse survival (HR > 1, Figure 10B). Relatively higher expression of these genes was noted primarily in MB Groups 3 and 4 (Figure 11).

The most significant differentially expressed SRG on 17q by the MB subgroup in the Cavalli dataset was *AXIN2* (Figure 11). Figure 12A shows that the expression of *AXIN2* was higher in the WNT subtypes than in all the other subtypes (F = 577.14, *p* < 1.0 × 10^−300^). Its expression was also higher than non-tumor samples in the Swartling dataset (*p* = 8.62 × 10^−127^). For *AXIN2*, high expression was associated with better survival (Figure 12B). Its hazard ratio was 0.64 (0.50–0.83, *p* = 6.09 × 10^−5^); elevated expression was associated with survival protection (Figure 12B). All individuals (N = 49) in the WNTα subtype had an elevated *AXIN2* expression, while 15/21 in the WNTβ had an elevated *AXIN2* expression.

The most significant SRG on 17q was *IGF2BP1* (HR = 1.34, *p* = 2.88 × 10^−8^). *IGF2BP1* is over-expressed in the Group 3γ MB samples and high expression was associated with a worse prognosis (Figure 13). In the Swartling meta-analysis [4], *IGF2BP1* was also over-expressed compared to that of non-tumor samples. Our GO enrichment analysis of 17q SRGs showed that *IGF2BP1* expression was associated with the significant GO terms mRNA transport and RNA localization.

Another 17q SRG associated with significant GO terms (Table 2) was *KPNB1* (Karyopherin). Its high expression was associated with worse survival (Figure 14) (HR = 1.40, *p* = 2.92 × 10^−5^). *KPNB1* expression was also specifically over-expressed in Group 3γ (Figure 14) in the Cavalli dataset and in Group 3 in the Swartling dataset. The *KPNB1* protein is involved in the nuclear import of proteins and also plays a role in mitosis [17,18].

## 3. Discussion

Although the SRGs were distributed on all chromosomes, our analysis showed over-representation of SRGs on chromosomes 6 and 17. Aberrations of chromosomes 6 and 17 in MB are well documented: in the Cavalli dataset monosomy 6 was found in 48 of the 49 WNTα subtype subjects, and isochromosome 17 was reported in MB groups 3 and 4 in the Cavalli report [3], with the greatest frequency in the Group 4β subtype (87 of the 109 subjects). Our analysis identifies SRGs whose expression is associated with these aberrations as well as some that are not. Results from the Cox proportional hazards regression analysis showed that 27 of the SRGs on chromosome 6 remained after removal of individuals with known SCNAs, whereas no significant number of chromosome 17 SRGs remained after removal of individuals with known SCNAs. We conclude that chromosome 6 contributes major SRGs unrelated to SCNAs such as monosomy 6. Because of the samples size reduction, we cannot rule out the possibility that some SRGs on chromosome 17 are unrelated to SCNAs such as isochromosome 17.

The SRGs we identified have some overlap with the Northcott driver genes [16] and 49 SRGs overlap with the top 1% of genes supporting the major molecular subgroups in the Cavalli study. However, the majority of SRGs on chromosomes 6 and 17 are not on the Northcott list of driver genes [16] and not in the top 1% of genes supporting the four major molecular subgroups in the Cavalli report. The four molecular subgroups and 12 subtypes reported, having been defined by unsupervised cluster analysis, do not necessarily correspond to the primary clinical outcomes, namely, overall survival. Zhu et al. [10] reported a twelve-gene signature as a prognostic tool to predict overall survival in MB. Two of the SRGs we identified on chromosomes 6 and 17, *SYNCRIP* and *EIF4A3*, overlapped with the twelve-gene signature reported by Zhu et al. [10] using the Cavalli dataset (referred to in their report as the Florence [sic] dataset). Different from the methods used by Zhu et al. [10], our survival analysis model adjusted for age but not for MB subgroup, thus, is more likely to identify SRGs which were expressed differentially by MB subgroups and subtypes. Herein, we discuss the most significant age-adjusted SRGs, located on chromosomes 6 and 17, and the biological processes associated with these genes. The SRGs allowed us to identify genes encoding proteins that should be further examined as potential therapeutic targets in selected subgroups or subtypes of MB.

The results from this study suggest that the expression of SRGs is partially explained by chromosomal aberrations. Monosomy 6 is associated with reduced gene expression in the WNT group, of many genes, including SRGs. Copy number aberrations of chromosome 6q have been used by Pfister et al. [19] in MB survival prediction. The gain of 6q contributed to a poor outcome, while 6q deletion was associated with better survival [19]. Aberrations of chromosome 17, including isochromosome 17, isodicentric 17, and loss of 17p, have also been used as prognostic factors in MB. Variations in expression of some of the SRGs would be expected to reflect these chromosomal aberrations.

### 3.1. Chromosome 6 SRGs

While monosomy 6, in one of the WNT subtypes, is the major chromosomal aberration reported for this chromosome, herein we discuss SRGs on 6p and 6q separately to facilitate the identification of biological processes and individual genes that are statistically associated with overall survival.

### 3.2. Chromosome 6p SRGs

Of all the 967 SRGs identified by the survival analysis, the most significant SRG was *HMGA1*, located at chromosome 6p21.31. It codes for a non-histone chromatin protein for many cellular processes, including cancer metastasis. As shown in Figure 5, a high expression of this gene in Group 3γ MB was associated with poor survival. *HMGA1* is a histone H1 competitor [20]. As such, it plays a role in the epigenetic regulation of gene expression.

Sumter et al. [21] reviewed the role of *HMGA1* as an oncogene for various tumors, including breast cancer, prostate cancer, and lung cancer, when over-expressed. GO enrichment analysis on chromosome 6 p SRGS has identified *HMGA1* as associated with the GO terms nucleosome organization and nucleosome assembly. This is consistent with its role as a histone H1 competitor. There are several additional MB SRGs in the nucleosome organization process. These include five genes in the histone cluster of genes located at 6p 22.1–22.2, expressed during the S-phase of the cell cycle [22], as well as *DAXX*, which encodes a histone linker, and *BRD2*, which encodes a histone transcription regulator. These data further support a key role for the nucleosome in Group 3γ MBs. Another SRGs in the nucleosome assembly GO term was *CENPW*, a gene located on chromosome 6q. It binds to nucleosomes at the centromere [23]. These data suggest the hypothesis that dysregulation of nucleosome components plays a role in MB survival related to gene expression of chromosome 6p. Our analysis highlights *HMGA1* and genes coding for nucleosome components as having a major significance in Group 3γ MB survival and as potential therapeutic targets in Group 3γ MBs. *HMGA1* and *DAXX* are both included in the ONGene database of oncogenes [24]. *CENPW* has been reported as a biomarker for hepatocellular carcinoma and a potential target for gene therapy in this cancer [23]. Our study suggests that *CENPW* is a marker for Group 4α MB and should be examined further as a potential therapeutic target for this MB subtype.

Other chromosome 6p SRGs whose expression was also elevated in Group 3γ include *SNRPC* (required for formation of the spliceosome), *XPO5* (involved in transport of small RNAs from nucleus to cytoplasm), *FANCE* (which encodes a protein that contributes to DNA cross link repair), and *H2BC8* (a gene located on the histone cluster of chromosome 6). These SRGs, as well as *HMGA1*, are on our list of chromosome 6 SRGs that remained after the removal of individuals with known SCNAs. From a statistical survival perspective, *HMGA1* was most significant.

Chromosome 6p amplification has been associated with various cancers by comparative genomic hybridization or FISH [25], leading to the hypothesis that these cancers are caused by an increased expression of one or more oncogenes on chromosome 6. While Cavalli et al. [3] showed some cases with chromosome 6p gain in Group 3γ, their report indicated that this is not a statistically significant gain compared to the other subgroups in Group 3 (see Figure 5E in the Cavalli study). To the best of our knowledge, significant 6p amplification has not been reported for MB. An alternative explanation for selectively increased SRG expression in Group 3γ is that it may represent an epigenetic phenomenon.

### 3.3. Chromosome 6q SRGs

*SYNCRIP* (aka HnRNP-Q), an 6q SRG with a high HR, is included in the list of ‘driver genes’ by Northcott et al. [16] as well as in the 12-gene signature for MB prognosis reported by Zhu et al. [10]. For several 6q SRGs with HR > 1, including *SYNCRIP*, expression was upregulated in Group 3 or Group 3γ (Figure 6). Since *SYNCRIP* expression was elevated in Group 3 MB compared to the non-tumor group at a high level of significance in the Swartling dataset (*p* = 2.01 × 10^−19^), overall, our study shows that, in addition to being on the Northcott list of driver genes [16], *SYNCRIP* is significantly related to the survival of Group 3 MBs. *SYNCRIP* encodes a splicing protein [26] and modulates mRNA translation [27] and transport [28].

*TCP1* (aka *CCT1*), another 6q SRG with an HR > 1, is located on 6q25.3. It encodes one of the proteins in the TCP1 ring complex, a molecular structure that folds proteins. Several SRGs coded for proteins contribute to the TCP1 ring structure. These SRGs, *TCP1* (Chr 6), *CCT2* (Chr 12), *CCT3* (Chr 1), *CCT4* (Chr 2), *CCT8* (Chr 21), were found, in the overall GO enrichment analysis (*p* < 0.01) of SRGs (Appendix A) and contribute to a highly significant group of GO terms related to the telomeric region of chromosomes and to the Cajal body, including regulation of telomerase RNA localization to the Cajal body. Kaplan-Meier analysis showed that high expression levels of *TCP1* (Chr 6), *CCT2* (Chr 12), *CCT3* (Chr 1), *CCT4* (Chr 2), *CCT8* (Chr 21) were associated with worse survival. Our analysis of these data suggests that dysregulation of the TCP1 ring complex is a significant factor contributing to survival of MB.

### 3.4. Chromosome 17 SRGs

Isochromosome 17, with loss of a copy of 17p and gain of 17q, is found in both Group 3 and Group 4 MB [3,29]. The isochromosome 17 aberration has been reported to be a prognostic factor in infant MB [30]. It has been suggested that isochromosome 17q may be a marker for uncontrolled cell proliferation in MB [31]. From our analysis, we conclude that a number of chromosome 17 SRGs are related to known SCNAs, primarily isochromosome 17.

Herein, we discuss the most significant SRGs on 17p and 17q in MB separately.

### 3.5. Chromosome 17p SRGs

The heatmap and cluster analysis of 17p SRGs illustrate that low expression is found in most of the Group 4β subjects (Figure 9). Worse survival with low expression of the genes in Figure 9 suggests the possibility of one or more tumor-suppressor genes in the list of SRGs on chromosome 17p. Our analysis suggests the interpretation that worse survival in Group 4β is due to a reduced expression of one or more SRGs with the loss of a copy of 17p, whether or not associated with the isochromosome 17 aberration. McCabe and colleagues reported that 17p loss predicated a poor prognosis [32] in pediatric MB. Cogen and McDonald [33] reviewed the reports of tumor suppressor genes on 17p in MB. They pointed out that the most common location for 17p loss was in the 17p13.1 cytogenetic band. We identified the 17p13.1-p13.3 region as the most frequently represented cytogenetic bands of the SRGs on chromosome 17. Our analysis supports the hypothesis that one or more tumor-suppressor genes among our list of 17p SRGs are specific for MB subtype Group 4β. While the tumor-suppressor gene *TP53* is located on 17p13.1, this gene was not found in our list of 17p SRGs.

The most significant 17p SRG in our model was *CYB5D2* (Cytochromes B5 domain containing 2). Low expression of this gene was associated with worse survival prognosis. *CYB5D2* was included in a 13-gene list of MB prognosis predictors [34]. Other cytochrome complex-related SRGs included *COX10* (Cytochrome C Oxidase Assembly Factor Heme), *SCO1* (Synthesis of Cytochrome C Oxidase), and *TTC19* (Tetratricopeptide Repeat Domain 19) (Table 1). Low expression of these three genes was associated with poorer survival. The lowest expression of these genes was noted in Group 4β. These data suggest a deficiency in the cytochrome oxidase C contributes to a worse prognosis in Group 4β.

One of the 17p SRGs, *CTDNEP1* (CTD Nuclear envelope phosphatase 1), was listed as an MB ‘driver’ gene by Northcott [16] and an oncogene by Luo et al. [35]. This gene is located at 17p13.1. High expression of this gene was associated with survival protection (HR = 0.72), while reduced expression of this gene in Group 4β was associated with worse survival. The GO enrichment analysis of 17p SRGs showed that *CTDNEP1* contributed to one of the major GO biological processes, nuclear membrane disassembly (Table 1). Reduced expression of *CTDNEP1* would be expected in cases with the loss of 17p, whether or not it is part of the isochromosome 17 aberration.

The finding that several 17p SRGs located at 17p13.1 and 17p13.3 contributed to the GO molecular process of telomeric DNA binding in the GO enrichment analysis of SRGs needs further elaboration. These genes include *SMG6* and *RPA1* (Replication Protein A 1), located at 17p13.3, and *CTC1* (CST Telomere Replication Maintenance Complex Component 1), located at 17p13.1. *SMG6* codes for a protein that is a part of the telomerase ribonucleotide complex and has a role in telomere regulation [36,37,38]. The RPA1 protein contributes to DNA replication, DNA damage response [39], and telomere maintenance [40,41]. *CTC1* (aka *C17orf68*) encodes a protein that inhibits telomere degradation and contributes to DNA damage repair [42,43]. Telomerase activation occurs in many cancers [44] and has been reported in childhood MB [45]. The telomerase inhibitor Imetelstat has been used in clinical trials for selective blood cancers [46]. It has also been used to inhibit the growth of MB cells injected into mice [47]. We suggest it should be investigated further in Group 3 MBs, particularly in those of the Group 3γ subtype.

Using the GO term Regulation of telomere maintenance, we queried all our identified SRGs using the R2 genomics platform and found eight that were significantly different among the four MB subgroups (*TCP1*, *CCT2*, *CCT3*, *CCT4*, *CCT8*, *SMG1*, *SMG6*, and *MAP3K4*). One of the eight SRGs related to telomere maintenance, *SMG6*, is located on chromosome 17p. Using the Kaplan–Meier analysis, high levels of *SMG6* were associated with protection, whereas low expression levels were associated with worse survival (HR = 0.73 with 95% CI 0.62. to 0.86). SMG6, a telomerase subunit, has been reported to bind to telomere DNA and contribute to telomere regulation [37,38]. Our analysis of these data suggests that dysregulation of the TCP1 ring complex and telomerase activity are major factors in the determination of survival in MB.

The GO analysis of 17p SRGs (Table 1) lists some biological processes that do not seem to be related to cancer, including the process of adult locomotory behavior. However, a search of the literature showed that a number of the genes associated with this process are, in fact, related to various cancers as well. *ARRB2* (β-arrestin 2) expression has been shown to modulate the growth of colorectal cancer [48], glioblastoma [49], lung cancer [50], ovarian cancer [51], and prostate cancer [52]. In the Cavalli dataset, high expression of *ARRB2* and *PAFAH1B1* was protective (HR = 0.70, *p* = 1.38 × 10^−5^ and HR = 0.72, *p* = 4.90 × 10^−5^, respectively).

### 3.6. Chromosome 17q SRGs

A gain in copy number of chromosome 17q has been reported to be associated with poor prognosis [13,19]. The heatmap of 17q SRGs (Figure 11) suggests that some of these genes were over-expressed in the Group 3γ subtype, and some in the WNT group. For a number of SRGs on chromosome 17q, high expression was associated with MB Groups 3 and 4. The high expression is consistent with isochromosome 17 aberrations. Targeting the proteins encoded by these SRGs could be a way of overcoming the negative effects of over-expression of these genes due to isochromosome 17.

*AXIN2* upregulation (at 17q24.1) was associated with survival protection (Figure 12) in the WNT subgroup. The AXIN2 protein is an inhibitor of the WNT signaling pathway [53], and a part of the destruction complex regulating β-catenin [54,55], which is an immunohistochemical marker for WNT tumors [11]. The AXIN2 protein also acts as a negative feedback signal limiting activation of the WNT pathway [53,56]. Mutations of *AXIN2* have been reported to activate the WNT pathway [57]. Our analysis shows that *AXIN2* over-expression, perhaps as part of 17q gain, is associated with better survival in the WNT group than in the other three MB groups. Over-expression of *AXIN2*, as part of the feedback loop, may contribute to survival protection by limiting the WNT pathway signaling.

Two 17q SRGs significantly over-expressed in Group 3γ were *IGF2BP1* and *KPNB1* (Figure 13 and Figure 14). For both, over-expression was associated with poor survival (HR > 1). To the best of our knowledge, there are no other reports linking these two genes to MB. However, *IGF2BP1* promotes tumor growth in various cancers [58]. IGF2BP1 enhances tumor growth by stabilizing mRNAs that code for cell cycle regulators [58,59]. The potential use of IGF2BP1 inhibitors in cancer therapy has been suggested by Huang et al. based on data showing that *IGF2BP1* is over-expressed in various cancers [60]. Our analysis suggests that the use of *IGF2BP1* inhibitors may be more useful in Group 3γ MBs than in the other subtypes.

*KPNB1* has been shown to stimulate proliferation of cancer cells in various cancers, including breast cancer [61], prostate cancer [62], gastric cancer [63], colon cancer [64], and ovarian cancer [65]. KPNB1 inhibitors have been reported to be effective in inhibiting the proliferation of cancer cells [18]. KPNB1 is involved in the transport of proteins and RNA to the nucleus [18,66]. Our GO enrichment analysis of 17q SRGs identified *KPNB1* as one of the genes associated with the mRNA transport process. We suggest that KPNB1 inhibitors be tested in Group 3γ cells as a step in determining its therapeutic potential in this subtype of MB. *KPNB1* has been identified as an oncogene in ovarian cancer [67]. Our analysis supports the hypothesis that *KPNB1* is a 17q oncogene that stimulates proliferation in Group 3γ MBs. In summary, various reports suggest that the KPNB1 protein may be a potential therapeutic target [17] in cancers. Our analysis suggests KPNB1 may be a potential therapeutic target in Group 3γ MBs. Our list of 17q SRGs may contain one or more oncogenes that are likely MB subtype-specific. It appears that, at least statistically, *KPNB1* fits the category. Its importance in MB is enhanced by the statistical evidence relating it to survival.

## 4. Methods and Materials

### 4.1. Data Sources

The Cavalli dataset: The gene expression dataset for 763 MB samples by Cavalli et al. (2017) [3] was downloaded from the R2 Genomics Analysis and Visualization Platform (Amsterdam, the Netherlands) (https://hgserver1.amc.nl, (accessed on 1 July 2024). The gene expression profile for each primary MB tumor sample was generated from the Affymetrix Human Gene 1.1 ST Array. Phenotypes, such as age, sex, MB subgroup, MB subtype, overall survival status, and survival years, were also included in the downloaded file.

The Swartling dataset: For some of the SRGs (see above, Figure 5, Figure 7, Figure 12, Figure 13 and Figure 14), we also presented their gene expression levels by the MB molecular subgroups using the data from Weishaupt et al. (2019) (i.e., the Swartling dataset in the R2 Genomics platform) [4]. This dataset contains normalized gene expression profiles from 1641 samples, including 1350 primary MB samples and 291 normal brain samples (cerebellum) from 23 transcription datasets [4]. The normal cerebellar tissues served as controls in this meta-analysis.

### 4.2. Data Analyses

For the Cavalli dataset, the Cox proportional hazards regression model was used to determine the relationships between the overall survival status (alive vs. not alive) and gene expression for each gene with age as the covariate. Survival year was used as the time variable. The function *coxph* from the R statistical package Survival was applied for this model. A gene was considered statistically significant at the Benjamini–Hochberg adjusted *p*-value < 0.05.

Age was selected as the covariate after we explored the relationship between survival years and age using Spearman’s correlation, the associations between survival years and sex, and overall survival status and age using the Wilcoxon rank–sum test. The association between overall survival status and sex was assessed using the Chi-squared test.

After identifying the SRGs, the genes were grouped by their chromosome and cytogenetic bands. The chromosomal location was determined with the gene symbol checker of the Human Genome Nomenclature Committee (HGNC) (Appendix A). Since the total number of protein-coding genes differs by chromosome, we expressed these data as a proportion of the number of SRGs to the total number of protein-coding genes for each chromosome. Using the two-sample z-test for the equality of two observed proportions, we tested if chromosomes 6 and 17 contained significantly more SRGs compared to the other chromosomes. The Cox proportional hazards regression model was also used to determine SRGs after the removal of individuals with known somatic copy number alterations (SCNAs) on chromosomes 6 and 17.

### 4.3. Kaplan–Meier and Pathway Analysis of SRGs

We presented heatmaps of gene expression profiles of the SRGs located on chromosomes 6 and 17 by molecular subgroup and subtype using the R2 Genomics Analysis and Visualization Platform. For the most significant SRGs, Kaplan–Meier curves were presented using the platform’s KaplanScan, which separates the individuals into high vs. low gene expression groups based on an optimum survival cut-off for a gene. Finally, GO enrichment analyses were conducted for all SRGs and those on chromosomes 6 and 17 using Cytoscape 3.8.2 with the ClueGO plug-in [68]. The following parameters were used with this application: Under the Ontologies/pathways setting, the GO-Biological Process was selected, with an adjusted Benjamini–Hochberg *p*-value cutoff of 0.05. Our list of SRGs was scanned for oncogenes using the list at https://www.oncokb.org/cancer-genes (accessed on 1 July 2024) and https://ongene.bioinfo-minzhao.org/ongene_human.txt (accessed on 1 July 2024).

## 5. Conclusions

We have identified chromosomes 6 and 17 as the location of over-representation of SRGs in the Cavalli dataset. We concluded that genes on chromosome 6 makes a major contribution to survival risk; a contribution, at least partially unrelated to monosomy 6. The most significant biological processes associated with these genes were the mitotic cell cycle and regulation of chromosome organization at the telomeric region.

Statistically, the most significant SRG was *HGMA1* on chromosome 6p. As a histone competitor, it functions as an important transcription factor and regulator of the nucleosome. High expression of this gene is found in the MB Group 3γ, which has the worst survival prognosis compared to the other MB subtypes. The most significant SRG on chromosome 6q was *SYNCRIP*, a gene previously described as an MB driver gene. Thus, high expression of selected chromosome 6 SRGs are markers of poor prognosis in the MB Group 3γ tumors and potential therapeutic targets for this MB subtype.

Eight SRGs (*TCP1*, *CCT2*, *CCT3*, *CCT4*, *CCT8*, *SMG1*, *SMG6*, and *MAP3K4*) contributed to the GO term regulation of telomere maintenance, including TCPI and MAP3K4, located on chromosome 6q and SMG6 on chromosome 17p. Five of these genes encoded components of the TCP1 complex, a structure in which protein folding occurs. The TCP1 complex has been shown to play a role in the regulation of telomerase [69]. The results from this study provide a rationale for clinical testing of the telomerase inhibitor Imetelstat in Group 3 MBs.

Our analysis supports the hypothesis that there are one or more tumor suppressor genes among our list of 17p SRGs. Included in the 17p SRGs is *CTDNEP1*, another gene on the Northcott list of driver genes. Decreased expression of this gene was associated with worse survival. Decreased expression of *CTDNEPI* was found in the MB Group 4β compared to all the 12 subtypes in the Cavalli dataset. *CTDNEP1* has recently been reported as a tumor suppressor in aggressive MB [35].

High expression of *AXIN2*, an SRG on 17q, was associated with survival protection in the WNTα subgroup. High hazard ratios were found for several SRGs at the telomere end of 17q. The over-expression of SRGs on 17q appears to be related to isochromosome 17 and poor prognosis. *KPNB1*, a 17q SRG and oncogene, over-expressed in Group 3γ MBs, encodes a protein over-expressed in various cancers. Several KPNB1 inhibitors are available [18] that could be used to determine their effectiveness as therapeutic agents in Group 3γ MBs.

Our analysis showed that the top genes supporting the four molecular groups in MB are not necessarily the genes most associated with survival. The SRGs identified in this study, however, provide information on potential therapeutic targets, some of which are MB subgroup- or subtype-specific.

## Figures and Tables

**Figure 1 ijms-25-07506-f001:**
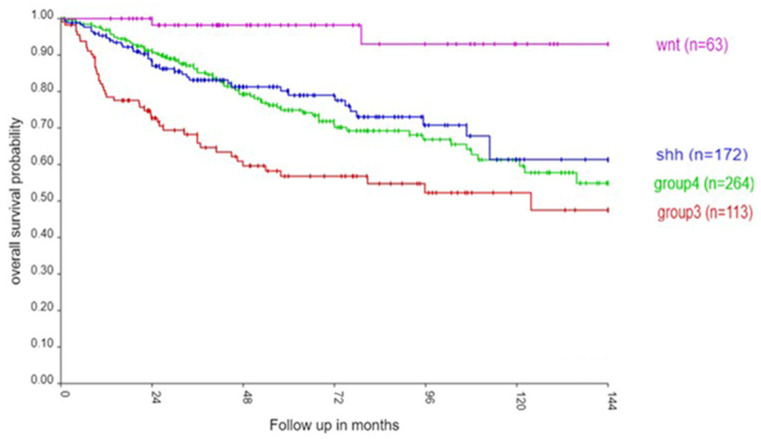
Kaplan–Meier curves for the four consensus MB subgroups in the Cavalli et al. (2017) dataset up to 144 months (χ2 = 31.68, *p* = 6.12 × 10^−7^). Sample sizes by MB subgroup: Group 3 (red) n = 113; Group 4 (green) n = 264; SHH (blue) n = 172; WNT (purple) n = 63.

**Figure 2 ijms-25-07506-f002:**
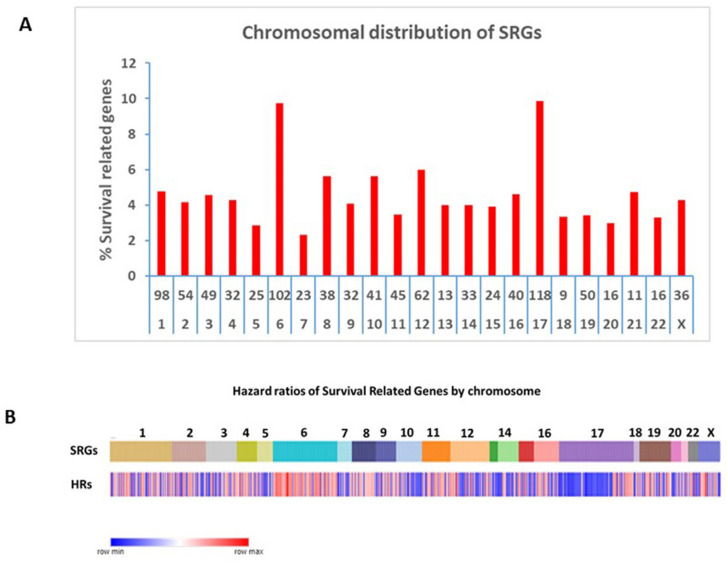
Chromosomal distribution of SRGs. (**A**) Distribution of the proportions of the number of SRGs to the total number of genes on each chromosome. For the x-axis legends, the top row shows the number of the SRGs, and the bottom row shows the chromosome number. The number of SRG genes for each chromosome was divided by the total number of protein-coding genes for each chromosome and expressed as a percentage in the y-axis legend. The highest proportions of the SRGs were noted on chromosomes 6 and 17 (both *p*-values < 0.0001), compared to the SRG proportions for the other chromosomes. (**B**) Chromosomal distribution of hazard ratios of SRGs. For chromosome 6, most of the hazard ratios for SRGs were >1 (red, i.e., high gene expression → poor survival). For chromosome 17, most of the hazard ratios for the p-arm SRGs were <1 (blue; i.e., high gene expression → better survival), while most of the hazard ratios for the q-arm SRGs were >1 (red; i.e., high gene expression → poor survival).

**Figure 3 ijms-25-07506-f003:**
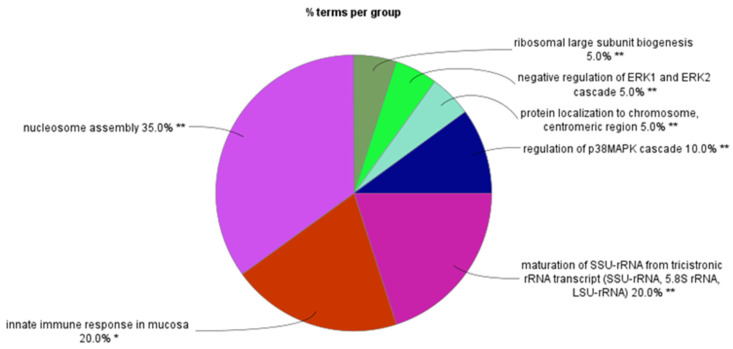
GO biological process enrichment analysis of SRGs on chromosome 6. Percentage of GO terms per group.

**Figure 4 ijms-25-07506-f004:**
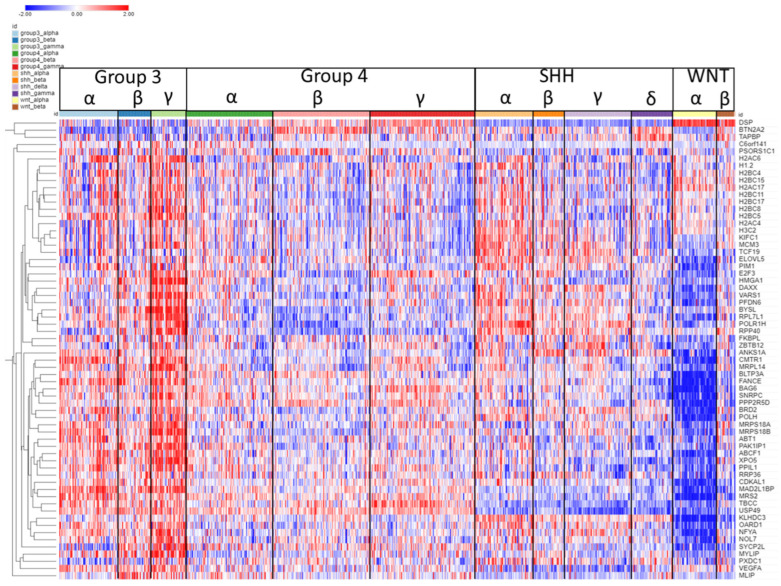
Heatmap and cluster analysis of the SRGs on chromosome 6p. Each column is an individual, each row is an SRG, and the color shows low (blue) to high (red) gene expression levels. The individuals were grouped by the MB molecular subgroups and subtypes. Increased gene expression with worse survival (HR > 1) is shown for 61/64 SRGs. Increased expression of the 6p SRGs (red) was found primarily in the Group 3γ subtype, while decreased expression (blue) was primarily in the WNTα subtype.

**Figure 5 ijms-25-07506-f005:**
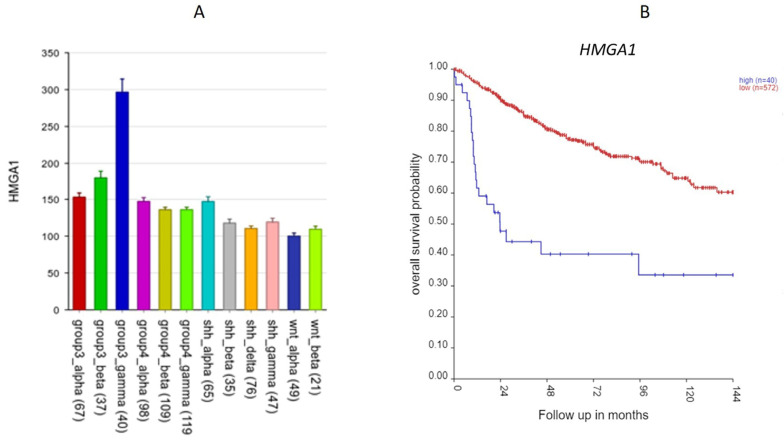
The *HMGA1* gene. (**A**) *HMGA1* expression by subtype. It was most elevated in the Group 3 gamma subtype (*p* < 0.0001). (**B**) Kaplan–Meier curves for the high (blue) vs. low (red) expression groups for *HMGA1* up to 144 months using the Cavalli dataset (*p* = 2.46 × 10^−11^, high n = 40, low n = 572).

**Figure 6 ijms-25-07506-f006:**
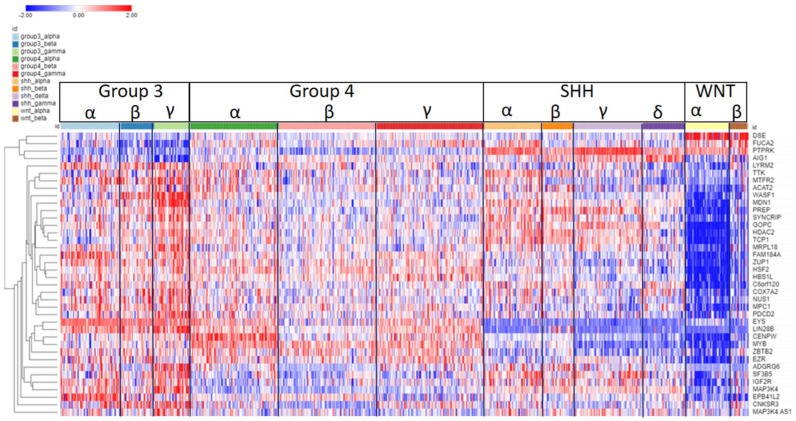
Heatmap and cluster analysis of the SRGs on chromosome 6q. Each column is an individual, each row is an SRG, and the color shows the low (blue) to high (red) gene expression levels. The individuals were grouped by the MB molecular subtypes and subgroups, and the genes were ordered by cluster analysis of expression.

**Figure 7 ijms-25-07506-f007:**
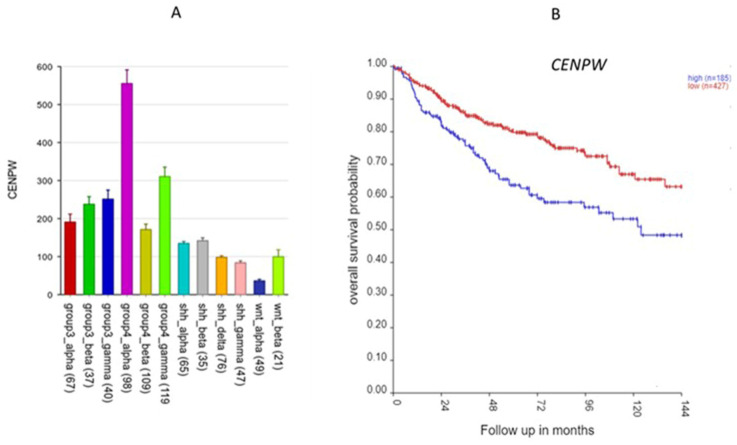
The CENPW gene. (**A**) Gene expression of *CENPW* by MB subtypes (*p* < 0.0001); (**B**) Kaplan–Meier curves (*p* = 2.25 × 10^−4^).

**Figure 8 ijms-25-07506-f008:**
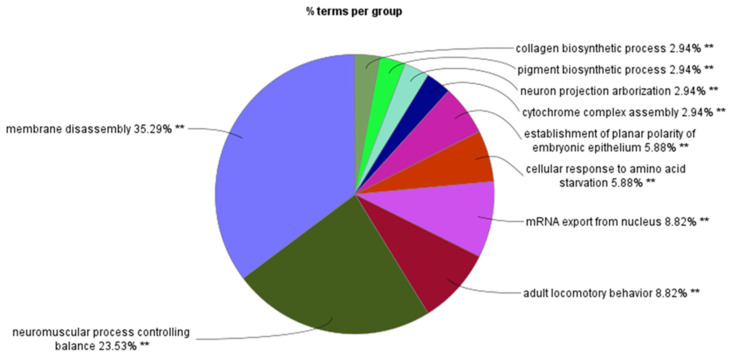
GO biological process enrichment analysis of SRGs on chromosome 17. Percentage of GO terms per group.

**Figure 9 ijms-25-07506-f009:**
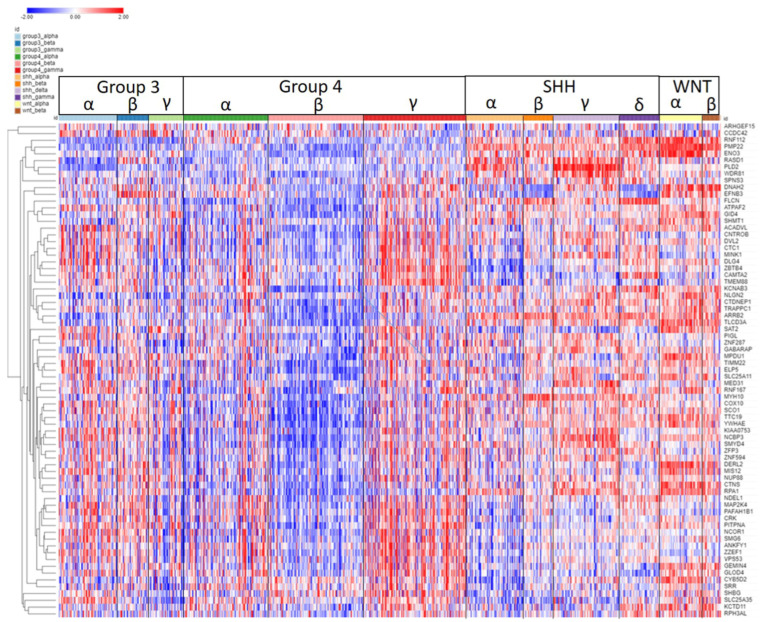
Heatmap and cluster analysis of the SRGs on chromosome 17p. Each column is an individual, each row is an SRG, and the color shows the low (blue) to high (red) gene expression levels. The individuals were grouped by the MB molecular subtypes and subgroups. Group 4β shows a relatively decreased expression of SRGs. Genes were ordered by cluster analysis of expression.

**Figure 10 ijms-25-07506-f010:**
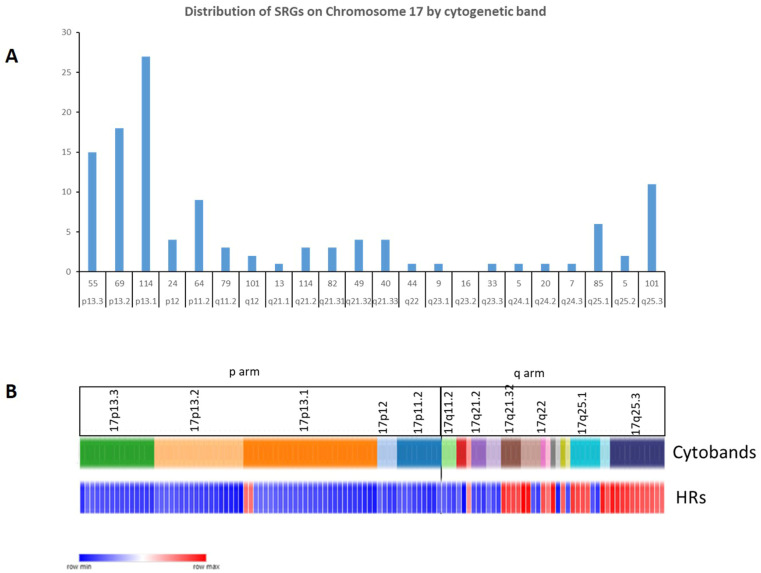
(**A**) The distribution of the SRGs on the p and q arms of chromosome 17 by cytogenetic band. The x-axis legend indicates the cytogenetic band and the total number of genes on each cytogenetic band. (**B**) Distribution of the SRG hazard ratios on chromosome 17 by cytogenetic band. Blue bars indicate HR < 1 for an SRG, while red bars indicate HR > 1. The p arm has more SRGs than the q arm. High hazard ratios are noted for SRGs at the telomere end of 17q.

**Figure 11 ijms-25-07506-f011:**
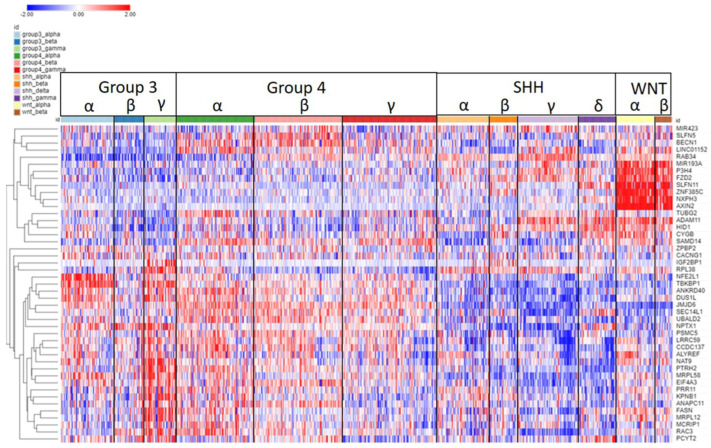
Heatmap and cluster analysis of the SRGs for MB on chromosome 17q. Each column is an individual, each row is an SRG, and the color shows the low (blue) to high (red) gene expression levels. The individuals were grouped by the MB molecular subtypes and subgroups. Genes were grouped by cluster analysis of expression.

**Figure 12 ijms-25-07506-f012:**
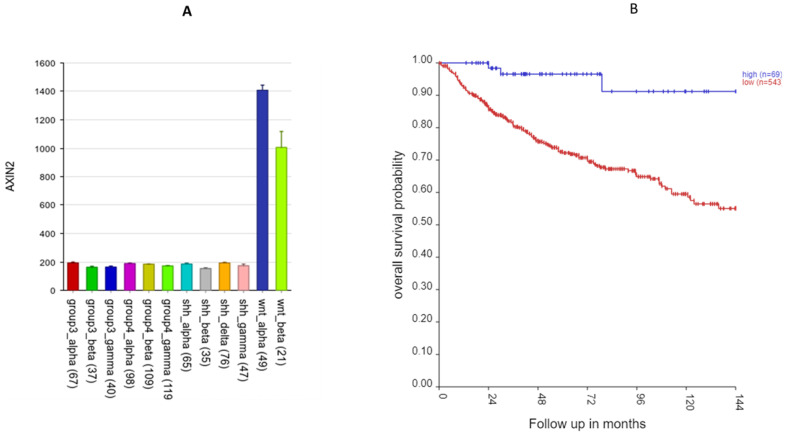
*AXIN2* on 17q. (**A**) *AXIN2* gene expression by MB subtype: higher expression in the WNT subgroup (F = 577.14, *p* < 1.0 × 10^−300^). (**B**) Kaplan–Meier curves: elevated expression of *AXIN2* is associated with better survival (*p* = 1.05 × 10^−4^).

**Figure 13 ijms-25-07506-f013:**
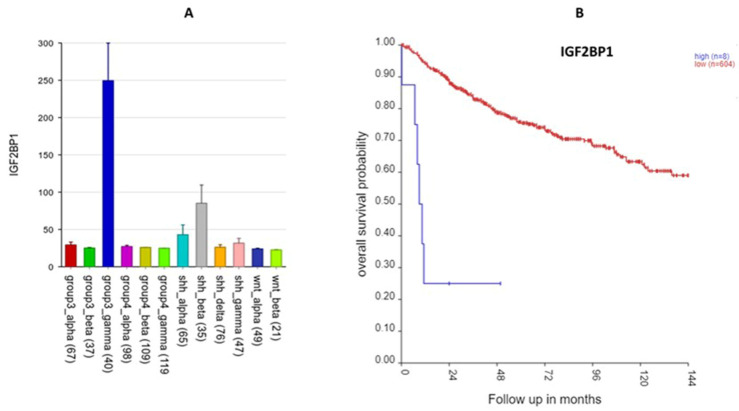
*IGF2BP1* on 17q. (**A**) *IGF2BP1* gene expression by MB subtype: higher expression in Group 3γ (F = 23.59, *p* = 6.81 × 10^−42^); (**B**) Kaplan–Meier survival curves: reduced expression of *IGF2BP1* is associated with better survival (*p* = 8.9 × 10^−12^).

**Figure 14 ijms-25-07506-f014:**
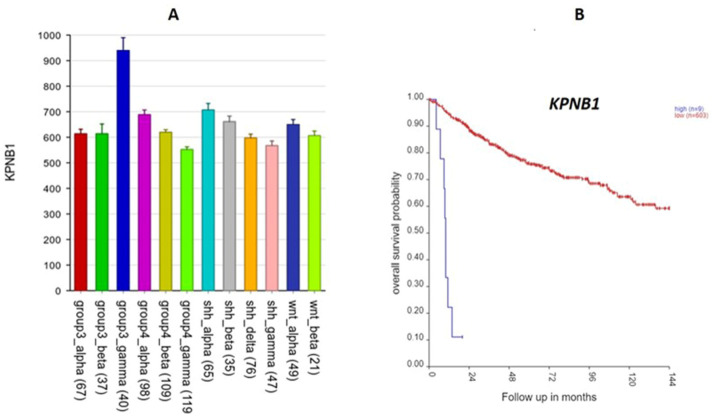
*KPNB1* on 17q. (**A**) Gene expression by MB subtype: *KPNB1* expression elevated in Group3γ (F = 20.27, *p* = 4.26 × 10^−36^). (**B**) Kaplan–Meier survival curves: elevated expression of this gene was associated with poor survival (*p* = 5.47 × 10^−26^).

**Table 1 ijms-25-07506-t001:** GO enrichment analysis of chromosome 17p SRGs.

Gene Ontology Enrichment Analysis of Chromosome 17p		
Gene Ontology Term	GO Term *p*-Value	SRGs
Nuclear membrane disassembly	6.53 × 10^−6^	*CTDNEP1*, *NDEL1*, *PAFAH1B1*
Adult locomotory (walking) behavior	1.37 × 10^−5^	*ARRB2*, *CTNS*, *EFNB3*, *NLGN2*, *PAFAH1B1*
Cytochrome complex assembly	3.12 × 10^−4^	*COX10*, *SCO1*, *TTC19*
Neuromuscular process controlling balance	1.03 × 10^−3^	*DLG4*, *NGLN2*, *PAFAH1B1*

**Table 2 ijms-25-07506-t002:** GO enrichment analysis of chromosome 17q SRGs.

Gene Ontology Term	GO Term *p*-Value	SRGs
Nuclear chromosome segregation	3.72 × 10^−6^	*ANAPC11*, *AXIN2*, *BECN1*, *HID1*, *KPNB1*, *P3H4*, *TUBG2*
RNA transport	4.56 × 10^−4^	*ALYREF*, *EIF4A3*, *IGF2BP1*, *KPNB1*

## Data Availability

These data referred to in this manuscript are publicly available at the R2 Genomics Analysis and Visualization Platform (http://r2.amc.nl (last accessed 1 July 2024) and at the NIH GEO database and are available upon reasonable request from the corresponding author.

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
