# Peer review of "Survival-Related Genes on Chromosomes 6 and 17 in Medulloblastoma"

_ijms, 2024, doi:10.3390/ijms25147506_

Round 1
Reviewer 1 Report
Comments and Suggestions for Authors
In the paper “Survival Related Genes on Chromosomes 6 and 17 in Medulloblastoma”, Vriend and Liu performed a meta-analysis with the Medulloblastoma (MB) transcriptome samples and identified several survival-related genes across different chromosomes. They also found chr6 and chr17 to be highly enriched of these SRGs compared to other chromosomes and also to be highly associated with Group 3 and WNT alpha subgroups of MB. To publish in the International Journal of Molecular Sciences, my specific comments are as follows:
- Please add the full wording of SRG before mentioning it as an acronym for the first time in the abstract section (Line 14).
- This work discusses the four major subgroups of MB based on molecular properties throughout the manuscript. However a thorough discussion regarding the subgroups is missing, which is needed for proper understanding of the described results.
- In this study, the authors primarily reported the role of chr6 and chr17 SRGs in Group 3 and WNT subgroup MBs. It would be interesting to see if the authors could find specific chromosomes responsible for other subtypes of MBs also (e.g. chromosomes responsible for the SHH subgroup; if any).
- Remove the placeholder text from Lines 65 to 67.
- In the Data Analyses section, the authors mentioned that for their bioinformatics analysis, they use ‘age’ as the covariate. The authors should consider stratifying their analysis result based on sex also, as studies have shown that gender-specific differences can be present in MB. It would be interesting to see if the authors could find sex-specific differences in SRGs.
- Fig. 2 has the wrong figure label: ‘Figure 2A’ (Line 142). Also, subfigure labels (a and b) are missing from this figure.
- The authors should consider performing some gene co-expression network analysis (and/or protein-protein network analysis) with the SRGs obtained from chr6 and chr17 to check how the genes in these communities are interacting with each other with possible regulatory roles.
Author Response
Response to Reviewer 1
We thank Reviewer 1 for the corrections and interesting comments.
- Comment: Please add the full wording of SRG before mentioning it as an acronym
Response: We have corrected the typo in line 11 (SRB) to SRG and included the full wording of SRG (survival related genes). SRG is now used in line 14.
- Comment: This work discusses the four major subgroups of MB…
Response: We have added a sentence describing the major signaling pathways in the SHH and WNT MB groups; currently there are no known signaling pathways that specifically characterize Groups 3 and 4 MB. The classification of medulloblastoma into 4 major subgroups goes back to a consensus agreement among scientists in the area and published in 2012 (Taylor et al).
- Comment: …It would be interesting to see if the authors could find specific chromosomes responsible for other subtypes of MBs also (e.g. chromosomes responsible for the SHH subgroup; if any)
Response.
The Cavalli manuscript (Cavalli et al., 2017) has a supplemental Table (Table S1) of copy number aberrations (gains and losses) for the various chromosomes. In our suppl. Table 1 we do list the chromosomal location of all the SRGs. We do not see a clear association of the SHH group with SRGS on a particular chromosome. A thorough analysis of SRGs on all chromosomes by MB subgroup and subtype could be done by analyzing copy number gains and losses, but that would result in a separate manuscript, probably as long as the present one.
- Comment: Remove the placeholder text from lines 65 to 67
Response: Done.
- Comment: In the Data Analyses section, the authors mentioned…they use ‘age as the covariate. …It would be interesting to see if the authors could find sex-specific differences in
Response:
We have added a sentence in the methods on the rationale for choosing age as the covariate:
‘Age was selected as the covariate after we explored the relationship between survival years and age using the Spearman's correlation, the associations between survival years and sex, and between overall survival status and age using the Wilcoxon rank-sum test. The association between overall survival status and sex was assessed using the Chi-squared test.
We have also added In the two sentences in the results on age and sex.
‘Our analysis revealed that age was significantly associated with survival years (p=0.02) but not with overall survival status (p=0.6). Sex was not significantly associated with either survival years (p=0.2) (Suppl. Fig. S1B) or overall survival status (p=0.55). Based on these findings, we selected age as a covariate in our survival analysis model.’
We have also added a graph showing the Kaplan-Meier curves for overall survival probability by sex in the supplemental figures, Figure 1B. A thorough analysis of this question of sex-specific differences in SRGs would be interesting but is beyond the scope of the present study. It would be another interesting project, considering that in the Cavalli dataset there is a predominance of male subjects.
- Comment; Fig 2 has the wrong label…
Response: We have corrected the labels for Fig.2 and added subfigure labels (a and b) to the figure
- Comment: The authors should consider performing some gene co-expression network analysis…
Response: It is not clear which program the reviewer has in mind. We did use Cytoscape for GO enrichment analysis. We did identify several genes making up the TCP1 ring complex and their relation to regulation of the telomere.
A protein-protein network analysis (or a transcription translation network analysis) would be great, but there would be problems finding a dataset for MB to compare to our SRGs.
A co-expression study using the R2 genomics and visualization platform could possibly be done for each of the SRGs of Chr 6 and Chr 17. That would require additional extensive and time consuming analysis and might not provide much more information than our current analysis. Another way of looking at gene co-expression would be to use factor analysis. Both of these approaches take us well beyond the scope of the present study.

Reviewer 2 Report
Comments and Suggestions for Authors
The manuscript from Jerry Vriend, et al, describes Survival Related Genes on Chromosomes 6 and 17 in Medullo-2 blastoma. The concerns are outlined below.
· Authors need to check the full form of survival related genes. Line 11, it referred as SRBs, and in line 56, it referred as SRGs.
· Line 32, full form of SHH, and WNT is missing
· Line 65, “This section may be divided by subheadings. It should provide a concise and precise description of the experimental results, their interpretation, as well as the experimental conclusions that can be drawn.”. It seems it is copied from manuscript template describing what needs to be included in methods sections. Do authors need to keep it for the main article?
· Line 74, survival related genes (see below)….., author may need to specify exactly where to see to avoid confusion to the readers.
· Line 87, SRG full form already mentioned in line 56, it is repeating here
· Ling 74, SRG abbreviation may be used for survival related genes
· Line 123, adjusting for age….. Authors may need to mention the criteria used for adjusting the age.
· Supplementary Table S1. Labeling is missing for chromosome column.
· Lin 124, pathway analysis… authors need to specify which pathway analysis they have performed. For instance, DAVID, Reactome, overrepresentation etc. The parameters used in Cytoscape software need to be elaborated in the methods section for reproducing the results.
· Line 124, “Pathway analysis showed that a total of 96 Gene On-124 tology (GO)”……. The supplementary table S2 showed 95 terms instead of 96.
· Line 131-132, the telomere and telomerase RNA localization to the Cajal body (Suppl. Table S1)….. Supple. Table 1 does not show any of these details.
· Line 156. “Figure 3 shows the GO pathways associated with SRGs on chromosome 6”…. However, Fig.3 represents % terms per group not pathways.
· Line 161, what are SRPs in Fig. 3 legends.
· Line 162, survival related genes.. in some places’ abbreviation used, for instance 3.1; and in some places, for instance 3.2 full form used. Need to be consistent entire manuscript.
· Line, 186, p-values may be represented like p = 8.36 x 10-13 instead of p = 8.36 x 10-13. The same applicable to other places, wherever p-value is represented.
· Authors showed Fig. 5B, but never described/cited in the main results section.
· Line 223-224, authors may need to cite Fig. 7a and b properly, instead of mentioned fig. 7.
· It is not clear whether authors wanted to describe pathways or biological processes category that they found in gene ontology enrichment analysis. For instance, Fig. 3 labeled as Pathway analysis of SRPs on chromosome 6. Percentage of GO terms per group. The fig represents biological processes but not pathways. Same applicable to Fig. 8. The data represents biological processes but is labeled as pathway analysis. Same applicable to Tabel 1 on page 10. Are authors referring GO enrichment analysis as pathway analysis?
· Line 315, …of 17q SRGs (Table 3).. Table 3 is missing.
· Line 414, “(see their Fig. 5E)”…. Fig. 5E is missing.
· Line 431, “overall pathway analysis (p < 0.01)of SRGs (Supplemental Table 1)…………..” Supplemental Table 1 doesn’t have pathway analysis. However, line 123 authors mentioned “Supplementary Table S1 lists the survival analysis of the SRGs by chromosome using the Cavalli dataset.”
· Authors may need to cross-check the usage of terminology for biological processes and biological pathways in the entire manuscript. For instance, Line 471, nuclear membrane disassembly referred as biological pathway though it comes under biological processes category in GO enrichment analysis. It seems authors referring all the biological processes as pathways.
· Line 527, is it KPBN1 or KPNB1? Same after line 527.
· In the supplementary figs word file, authors mentioned the legend of Supplemental Figure 3. But there is no figure.
· Line 590, Video S1: title… There is no video file.
Author Response
Response to Reviewer 2
We thank Reviewer 2 for the review
Line 11. Authors need to check the full form of survival related genes. Line 11, it is referred as SRBs, and in line 56, it is referred as SRGS.
Response: SRBs corrected to SRGs
Line 32. Full form of SHH and WNT is missing.
Response: Full form of SHH and WNT provided
Line 65. …It seems it is copied from manuscript template…
Response: Placeholder lines deleted
Line 74. Author may need to specify exactly where…
Response: specific figures indicated
Line 74, 87, SRG full form already mentioned.
Response. Ok full form deleted
Line 123, adjusting for age…authors may need to mention the criteria used for adjusting age.
Response. We have added two sentences in the methods on the rationale for choosing age as the covariate:
‘Age was selected as the covariate after we explored the relationship between survival years and age using the Spearman's correlation, the associations between survival years and sex, and between overall survival status and age using the Wilcoxon rank-sum test. The association between overall survival status and sex was assessed using the Chi-squared test.’
In the Results we have added the following:
Our analysis revealed that age was significantly associated with survival years (p=0.02) but not with overall survival status (p=0.6). Sex was not significantly associated with ei-ther survival years (p=0.2) (Suppl. Fig. S1B) or overall survival status (p=0.55). Based on these findings, we selected age as a covariate in our survival analysis model.
Comment: Supplementary Table 1 missing chromosome column.
Response. Heading for column added.
Comment. Line 124, pathway analysis. Authors need to specify which pathway analysis they have performed…..The parameters used in cytoscape software need to be elaborated…
Response. The following parameters were used with this application: Under the Ontologies/pathways setting Go-BiologicalProcess was selected, with an adjusted Benjamini-Hochberg p value of 0.05. This information was added to the Methods.
Comment. Line 124 showed a total of 96 genes. The supplementary table S2 showed 95 terms.
Response. Corrected to 95
Line 131-132. The telomere and telomerase RNA localization to the Cajal body (Suppl. Table S1)…Suppl. Table S1 does not show any of these details.
Response. I have corrected the error. The information is found in Suppl. Table S2.
Line 156. Figure 3 shows the GO pathways associated with SRGs on chromosome 6. However, Fig. 3 represent % terms per group not pathways.
Response. I have changed the legend of Fig.3 to ‘GO Biological Process enrichment analysis of SRGs on Chromosome 6’. Also changed for Fig. 8
Line 161. I have corrected the typo. SRPs to SRGs.
Line 162. In some places abbreviation for survival related genes used, and in some places the full form used. Need to be consistent.
Response. I have replaced the full form with the abbreviation throughout the manuscript.
Line 186. P-values may be represented like p = 8.36 x 10-13… I have changed the exponents accordingly, throughout the manuscript.
Comment. Authors showed Fig. 5B, but never cited it in the main results section
Response. I had add a sentence: ‘Fig. 5B shows that high expression of HMGA1 was associated with poor survival.’
Line 223.224, authors may need to cite Fig 7A and 7B properly.
Response. I have now cited Fig 7A and Fig7B separately.
Line 221-222 I have changed ‘nucleosome assembly pathway’ to nucleosome assembly biological process’.
I have changed the title of Table 1 on page 10 from ‘Gene ontology analysis of chromosome 17p’ to ‘Gene ontology enrichment analysis of chromosome 17p’.
Line 315. Table 3 is missing.
Response. Table 3 is not required. I have deleted the reference to it.
Line 414. (see their Fig. 5E). Fig. 5E is missing
Response. We are referring to a figure in the Cavalli publication. I have added the clarification.
“(see Fig. 5E in the Cavalli reference). It is not one of the figures of our manuscript.
Line 431. Supplemental Table 1 doesn’t have pathway analysis.
Response. I have corrected it to Table 2. Table 2 has the GO enrichment analysis
Line 471. I have changed ’biological pathway’ to GO ‘biological processes’.
Line 527. I have corrected the spelling of KPNB1.
Comment. Authors need to cross-check the usage of terminology for biological processes and biological pathways in the entire manuscript.
Response. In a number of cases throughout the manuscript we changed biological pathways to biological processes.
We have indicated the changes in red in the revised manuscript.
